# Mating Behavior and Identification of Male-Produced Pheromone Components in Two Woodwasps, *Sirex noctilio* and *Sirex nitobei*, in China

**DOI:** 10.3390/insects13100966

**Published:** 2022-10-21

**Authors:** Pengfei Lu, Enhua Hao, Min Bao, Rui Liu, Ciyuan Gao, Haili Qiao

**Affiliations:** 1The Key Laboratory for Silviculture and Conservation of the Ministry of Education, School of Forestry, Beijing Forestry University, Beijing 100083, China; 2Institute of Medicinal Plant Development, Chinese Academy of Medical Sciences and Peking Union Medical College, Beijing 100193, China

**Keywords:** *Sirex noctilio*, *Sirex nitobei*, Siricidae, mating, pheromone, (*Z*)-3-decenol

## Abstract

**Simple Summary:**

The woodwasp *Sirex noctilio* Fabricius is a major quarantine pest that was first discovered in China in 2013 and mainly harms *Pinus sylvestris* var. *mongolica*. *Sirex nitobei* Matsumura is native to China and closely related to *S. noctilio*. Recently, *S. nitobei* and *S. noctilio* were found to be attacking *P. sylvestris* var. *mongolica* from June to September successively in China. In recent years, pheromones have been useful tools for the development of pest evaluation and management strategies. Therefore, the aim of this study was to investigate the characteristics of two species of wood wasps (*S. noctilio* and *S. nitobei*): (1) mating behavior and circadian rhythms; (2) the factors influencing mating behavior; (3) the male-produced pheromone components in both species based on three reported components from *S. noctilio*. Finally, what we found was that most of the two species of wood wasps mated at two-days-old; within the same day, mating behavior peaks between 11:00 and 12:00; light conditions, individual size, and female–male proportion affect their mating behavior; (*Z*)-3-decenol became the main component of pheromone of both species. By investigating the mating behavior of two species of wood wasps and analyzing the pheromone production, we could proceed targeted control measurements during the mating period of wood wasps and provide strategic significance for the subsequent control.

**Abstract:**

To protect vulnerable trees from native and invasive wood wasps, the mating behavior of these two woodwasp species (*S. noctilio* and *S. nitobei*, respectively) and factors influencing this behavior were investigated in cages outdoors. Male-produced pheromones were identified in both woodwasp species. Compared with the native species *S. nitobei,* the invasive species *S. noctilio* showed stronger mating ability, including mating frequency, time, and duration. The mating behavior of both species mainly occurred from 9:00 to 17:00 each day, peaking at 11:00 and 12:00. The daily mating behavior of both species was most directly related to light intensity. Both female and male *S. noctilio* and *S. nitobei* were capable of mating upon emergence, and most individuals mated at 2 days of age. For both species, a female-to-male ratio of 5:15 was most conducive to mating, and individuals with a larger body size were preferred as mates by males and females. (*Z*)-3-decenol was present in solid-phase microextraction extracts of both species. Two reported minor reference components, (*Z*)-4-decen-1-ol and *(E, E*)-2,4-decadienal, were not identified in either woodwasp species. The peak of male pheromone release occurred from 11:00–12:00 for 2-day-old individuals.

## 1. Introduction

Woodwasps (Hymenoptera: Siricidae) are important hymenopterans that seriously damage various species of coniferous and hardwood trees. Female wasps generally attack weakened and stressed trees, and they may lay up to 400 eggs in the outer sapwood of their hosts [1]. Females carry arthrospores of a specific fungus, *Amylostereum* spp., as well as phytotoxic mucus in specialized internal organs called mycangia. Both fungus and mucus are synchronously injected into oviposition holes through the bark along with eggs [2]. The fungus produces enzymes capable of decaying the wood around larval tunnels, providing a convenient food source for larval feeding. Eventually, the fungus, phytotoxic mucus, and wasps work together to kill the chosen host tree.

*Sirex noctilio* Fabricius are a kind of destructive woodwasp that attack trees in their native habitats and they have been introduced as invasive pests around the world. *S. noctilio* is native to southern Europe, North Africa, and parts of Asia, but it has been an invasive species in the Southern Hemisphere since the early 1900s, as well as in North America since 2005 [3,4]. *S. noctilio* was introduced into China in Daqing, Heilongjiang Province, in 2013, after which it spread to Liaoning, Jilin, and Inner Mongolia in northeastern China, where it reportedly attacks only *Pinus sylvestris* var. *mongolica* Litv. [5,6]. *S. nitobei* is distributed throughout Japan, with the exception of Hokkaido Island, as well as in North Korea and China (Fukuda et al. 1993). In China, the species is native and mainly distributed in the provinces of Shanxi, Yunnan, Inner Mongolia, and Hebei [7]. *S. nitobei* poses a hazard to ancient and debilitated pines, such as the *Pinus tabuliformis* in Xiangshan Park in Beijing, and is a significant threat to other pine species such as *P. armandi* and *Larix* spp. [8].

The morphology of *S. noctilio* and *S. nitobei* is very similar, with the most obvious differences being the differing colors of the abdomen and hindfoot. In China, both species are associated with the same fungal symbiont, *Amylostereum areolatum* [5,9]. Recently, *S. noctilio* and *S. nitobei* were found to be attacking *P. sylvestris* var. *mongolica* from June to September from 2016 to 2018 successively, in Jinbaotun town, Inner Mongolia Autonomous Region [10]. *S. noctilio* adults started to emerge in the field from late June to early September; subsequently, *S. nitobei* emerged from the same trees, with a peak emergence rate from late August through late September. In order to inhibit the spread of these woodwasps and prevent them from inflicting damage upon trees, integrated pest management strategies are urgently needed, but the development of such strategies requires detailed knowledge of the biology and behavior of these wasps, including the cellular mechanisms underlying mating and host selection, as well as the environmental factors influencing such behavior.

As an early invasive species, the mating and oviposition behavior of *S. noctilio* has been studied in South Africa, New Zealand, and South America [11,12]. However, immediately following emergence, males consistently flew up to the crown of the tree upon which they had emerged [2]. Therefore, mating took place on the upper branches of trees, making it difficult to observe how female and male siricids found each other in the wild [13]. In a recent study by Caetano and Hajeck [14], the mating behavior of *S. noctilio* was observed in a laboratory setting. However, as an invasive species, the in situ mating behavior of *S. noctilio* on *P. sylvestris* var. *mongolica* in China and factors influencing such behavior are important considerations for the development of pest control strategies, yet they remain poorly understood.

Although *S. nitobei* is native to China, it is relatively understudied because of its low abundance and habit of attacking only dead or highly stressed trees with little economic value. However, recently, *S. nitobei* has attracted attention due to its sympatric coexistence with the quarantine pest *S. noctilio*. Takao et al. [15] found that the life cycle of *S. nitobei* was usually completed in one year, and it was reported that the species may have been more responsible for the death of Japanese pine trees than had previously been considered up to that point. Fukuda et al. [16] found that the oviposition success of *S. nitobei* was related to host tree conditions and its body size. Fitza et al. [17] showed the mutualism relationship between *S. nitobei* and *Amylostereum* fungus symbionts.

The spread of *S. noctilio* as an invasive species necessitates the development of a monitoring technique in newly invaded habitats. For example, trap trees treated with herbicide or girdling have been used to monitor and survey *S. noctilio* populations [18]. Kairomone (plant volatiles) lure traps were found to be the most effective method for trapping *S. noctilio* individuals in areas where populations were large [19]. Pheromones are commonly used as attractants to evaluate pest populations, and several pheromone compounds for *S. noctilio* are known [13,20]. Female pheromones derived from cuticular washes were identified as (*Z*)-7-heptacosene, (*Z*)-7-nonacosene, and (*Z*)-9-nonacosene; these pheromones were active in laboratory assay experiments and posited to be short-distance contact pheromones [20]. A male-produced pheromone was identified as (*Z*)-3-decen-1-ol and it was found to produce strong antennal response in males and females in gas chromatography-electroantennographic detection (GC-EAD) assays. In addition, (*Z*)-3-decen-1-ol was found to be the most active substance in Y-tube olfactometer and wind tunnel assays. However, several unknown or suspected minor components, including (*Z*)-4-decen-1-ol and (*E, E*)-2,4-decadienal, were not identified by GC-MS and only active in behavioral tests [13]. In contrast, no pheromones have been reported for *S. nitobei*. However, as congeneric species, *S. noctilio* and *S. nitobei* might utilize similar or identical pheromones, therefore, previously identified pheromones with efficacy in *S. noctilio* represent a rational starting point for an exploration of compounds with behavioral effects in *S. nitobei*. Once such pheromones are identified, they have the potential to be used as effective synergistic components in the development of plant-derived lures.

In woodwasps such as *S. noctilio* and *S. nitobei*, pheromone release is closely related to mating behavior, and pheromones are useful tools for the development of pest evaluation and management strategies, which are urgently needed to protect vulnerable trees from woodwasp infestation in native and invaded habitats. Therefore, the objectives of this study were: (1) to investigate the mating behavior and circadian rhythms of *S. noctilio* and *S. nitobei* in China; (2) to analyze the factors influencing mating behavior in these two species; and (3) to identify male-produced pheromone components in both species based on three reported components from *S. noctilio*. The goal of this study was to provide useful basic biological information to researchers developing integrated pest management strategies, and to those developing monitoring strategies in particular.

## 2. Materials and Methods

### 2.1. Insects

Infested *P. sylvestris* var. *mongolica* were felled, cut into bolts of about 1 m in length, and sealed at the ends with paraffin in early May 2019, which was several months prior to adult emergence. Next, bolts with pores on the trunk were taken from the field into a quarantine room at Jinbaotun Forestry Station (JFS) in Inner Mongolia Autonomous Region, China (39°58′ N, 116°13′ E), and kept in a fine nylon mesh cage (3 m length, 3 m width, and 3 m height) until adult emergence. The average environmental conditions for log storage were approximately 24 ± 1 °C and 65–70% RH. The cages were monitored daily until siricid emergence.

After emergence, adult wasps were sexed and then placed individually in smaller cages (20 × 20 × 20 cm) with a piece of filter paper inside. The age of each adult was recorded. Subsequently, adults were transferred into other rooms with environmental conditions similar to those mentioned above. Cages containing individuals of the same sex were kept in each room, and exposure to direct sunlight was avoided. Prior to the mating behavior experiments, the wasps were not allowed to mate and were not exposed to any synthetic odor source.

Adults emerging between 0 and 24 h were considered to be 1-day-old individuals. Each wasp was used only once. The wasps used for mating observation were 1–7 days old.

### 2.2. Mating Behavior Observation and Circadian Rhythm

Upon emergence, active 2-day-old virgin wasps (11 female and 33 male) were introduced into a single cage (45 × 65 × 135 cm) to mate. The experiment was replicated three times (three cages). Observation began immediately after pairing. The daily mating rhythm was recorded every 20 min from 8:00 to 18:00. The onset time of mating and mating duration were recorded for each pair. The cumulative average number of pairs that had begun to mate in each cage was calculated once per hour and subjected to data analysis. Once per hour, light intensity was recorded by a TES1330A illuminometer (Tes Electrical Electronic Corp., Taipei, Taiwan, China), while temperature and humidity were recorded by a DeLi 9010 thermometer (Ningbo Deli E-Commerce Co., Ltd., Ningbo, China) The experiment was conducted in a quarantine room at JFS. During each observation, deceased individuals were removed from the cage and replaced with new 2-day-old active subjects.

### 2.3. Effect of Adult Body Size on Mating Success

After the trials described above, all mated females and males were collected, and their body length, thorax width, proala length, and ovipositor length were measured using a Vernier caliper (Pro’s Kit PD-151 Digital Caliper, Shenzhen Jin Tao Rui Technology Co., Ltd., Shenzhen, China).

### 2.4. Effect of Adult Age on the Mating Frequency

The effect of age on the mating frequency of female and male adults was studied. To test the effect of female age on mating frequency, 1-day-old females (N = 11) and 2-day-old males (N = 33) were introduced into a cage (45 × 65 × 135 cm) and observed continuously for 7 days. The age of the female subjects was gradually increased, but males were replaced with 2-day-old active wasps every day to ensure that their age remained constant. The same arrangement was applied to test the effect of male age on mating frequency, but the sexes of the retained and replaced groups were switched, i.e., females were replaced daily with 2-day-old active subjects, and males were allowed to age during the experiment. Three replicates (3 cages) were performed for each treatment. Observation began immediately after pairing. The observation was conducted continuously, and we count the cumulative average number of pairs during the whole 20 min from 10:00 to 13:00, which was the peak mating period. The cumulative average number of pairs was calculated once per hour and subjected to data analysis.

### 2.5. Effect of Sex Ratio on Daily Mating Frequency

After emergence, 2-day-old virgin females and males were introduced into the copulating cage (45 × 65 × 135 cm) at different female: male ratios (5:5; 5:10, 5:15, 5:20) with 3 replicates (3 cages) for each ratio. Observation began immediately after pairing. Mating was recorded every 20 min from 10:00 to 13:00, which was the peak mating period. The cumulative average number of pairs that had begun to mate per cage was calculated once per hour and subjected to data analysis.

### 2.6. Headspace Collection of Male-Produced Pheromones

Volatiles were collected by headspace solid-phase microextraction (SPME) using a polydimethylsiloxane fiber (film thickness 100 µm; Supelco Inc., Bellefonte, PA, USA). A group of 15 virgin males (2-day-old) with average body weight was placed inside a sterile 500 mL tissue culture flask with a ventilation cap (Sarstedt, Leicester, UK). The SPME fiber was conditioned before use by baking it for 2 min at the injection port of a gas chromatograph (GC) at 270 °C, after which it was inserted through the cap and exposed to the sample headspace. Direct contact between the fiber and the wasps was prevented by a fine net surrounding the needle. Volatiles were collected from 8:00 to 18:00 at 1 h intervals (15 males). Three experimental replicates with different individuals (15 × 3 males) were used for each of the ten sampling time-points (once per hour from 8:00 to 18:00).

Age-dependent pheromone release was measured using virgin males aged from 1 to 7 days. For each test day, a group of 15 virgin males with average body weight was placed inside a sterile 500 mL tissue culture flask with a ventilation cap (Sarstedt, Leicester, UK). Volatiles were collected from 11:00 to 12:00 (15 males). For each of the seven groups (1, 2, 3, 4, 5, 6, or 7 days old), three experimental replicates (15 × 3 males) with different individuals were used for each sampling time point.

### 2.7. Gas Chromatography/Mass Spectrometry (GC/MS) Analysis

The SPME fibers were thermally desorbed into a splitless GC injector at 270 °C for 2 min. The GC/MS system consisted of an Agilent Technologies 5977A MS (Agilent, Santa Clara, CA, USA) coupled to an Agilent Technologies 7890 B GC (Agilent). For the HP-5 fused-silica column (30 m × 0.25 mm ID, 0.25 µm film, J&W Scientific Inc., Folsom, CA, USA), the initial oven temperature was maintained at 40 °C for 2 min, then increased at 6 °C/min to 180 °C, then increased at 15 °C/min to 270 °C. For the DB-WAX column (30 m × 0.25 mm ID, 0.25 µm film, J&W Scientific Inc., Folsom, CA, USA), the initial oven temperature was maintained at 50 °C for 1 min, then increased at 5 °C/min to 240 °C and maintained for 10 min. The NIST Mass Spectral Search Program (version 1.7) was used for data analysis. Injections were made in splitless mode. Helium was used as the carrier gas (1.0 mL/min). For electron impact (EI) mass spectra, the ionization voltage was 70 eV, the temperature of the ion source was 250 °C, and the temperature of the interface was 250 °C. Assignment of chemical structures to chromatographic peaks was based on comparison of their mass spectra fragmentation patterns and retention indexes with those of authentic standards. Circadian and age-dependent pheromone release was achieved by quantities expressed relative to the most abundant (*Z*)-3-decen-1-ol release period (set to a value of 100) during 11:00–12:00 in 2-day-old individuals for the two species.

### 2.8. Chemicals

The sex pheromone compound (*Z*)-3-decen-1-ol was purchased from Shuiguang Technology Co., Ltd. (Beijing, China). (*Z*)-4-decen-1-ol and (*E, E*)-2,4-decadienal was purchased from Sigma-Aldrich Co. (St. Louis, MO, USA). n-Hexane was redistilled before use. The compounds were found to be 98% pure by GC analysis.

### 2.9. Statistical Analysis

The effect of the time of day on mating frequency, the effects of environmental factors at different times of the day, the effect of adult age on mating frequency, the effect of the sex ratio on mating, circadian and age-dependent pheromone release were analyzed by ANOVA. Tukey’s multiple range test was used to identify significant differences between means. Student’s paired-sample *t*-test was used to identify significant differences between body size of mated and unmated siricids. Correlational analysis was performed to assess the strength of the relationships between environmental parameters and mating behavior. Significant differences in mating circadian rhythms, effects of age and sex ratio on mating frequency between two siricids were analyzed by Mann–Whitney *U*-tests. All data were analyzed with SPSS v. 16.0 (IBM Co., Armonk, NY, USA).

## 3. Results

### 3.1. Mating Behavior Observation

Similar mating behavior was observed for both woodwasp species. Compared with the native species *S. nitobei,* the invasive species *S. noctilio* showed stronger mating ability, including mating frequency and mating duration. *S. noctilio* cumulatively mated up to 19 and 13 times per day for female and male individuals, respectively, and the average mating duration was 51.25 ± 25.54 s for each mating session. Female *S. nitobei* cumulatively mated up to three times per day, whereas each male mated at most four times per day, and the average mating duration was 32.19 ± 16.02 s for each mating session. In general, *S. nitobei* males were more active and contacted females directly without additional triggering signals from females; however, male *S. noctilio* needed to touch the females using their antennae to gain agreement for mating. The mating procedure was divided into five phases: searching, attracting, seizing, copulating, and ending.

After being released, all of the males quickly gathered toward the top corner of the cage, where the sunlight was strongest. Some females remaineded still and others walked randomly along the cage walls, after which they slowly gathered in the corner occupied by the males. Most mating behavior occurred on the cage walls toward the sun. Occasionally, both females and males dropped from the wall and encountered on the cage floor, where they also mated. No mating behavior was observed during flight in the cage.

Searching: Males quickly gathered in the corner with direct sunlight after they were released, and females gradually gathered in the same corner soon after, allowing the males to find the females (Figure 1A(a); Figure 1B(a)).

Attracting: Females began to attract males when they were within approximately 1–3 cm. The attracted male *S. noctilio* walked tentatively to the females and touched them using their antennae (Figure 1A(b)). In contrast, male *S. nitobei* were more active, did not require an attracting signal from females, and always contacted females directly (Figure 1B(b)).

Seizing: After a preliminary test, the male would climb onto the dorsal part of the female from one side with his front legs. The male grasped the thorax of the female with his front and middle legs and fixed the posterior abdomen of the female with his hind legs. Meanwhile, the male attempted to curve his abdomen inward and mate (Figure 1A(c); Figure 1B(c)). In some cases, when the female refused to mate, she fanned her wings and escaped immediately. Males were observed to give up after several failed attempts to mate.

Copulating: Once a male was eventually accepted, mating occurred (Figure 1A(d,e); Figure 1B(c,d)). During the mating process, both males and females remained stationary. Sometimes the mating occurred in the opposite orientation after rotation of the male by 180 degrees (Figure 2d).

Ending: Mating ended when the genitals of the male and female were separated (Figure 1A(f); Figure 1B(e,f).

### 3.2. Mating Circadian Rhythms of Two Siricids

A similar mating rhythm was observed for both species of woodwasp. Mating mainly occurred between 11:00 and 12:00, with a significant difference between this period and the other time periods in *S. noctilio* (*P_S. noc_* < 0.001, *F_S. noc_* = 25.778, *df* = 9) and *S. nitobei* (*P_S. nit_* < 0.001, *F_S. nit_* = 13.184, *df* = 9). No mating behavior was observed before 9:00 or after 17:00 (Figure 2). Both species mainly mated on the wall of the cage and seldom on the ground. Compared with *S. nitobei,* the average mating frequency of *S. noctilio* was much higher during each 1 h test period.

The daily illumination and temperature rose to a peak at 11:00–12:00 and then gradually decreased (*P_illu_* = 0.091, *F_illu_* = 2.105, *df* = 9; *P_temp_* = 0.535, *F_temp_* = 0.903, *df* = 9). The daily humidity reached its lowest point at 11:00–12:00 and then gradually increased (*P_humi_* = 0.714, *F_humi_* = 0.666, *df* = 9) (Table 1).

The correlational analysis between the selected environmental parameters and mating behavior indicated that illumination was most strongly associated with mating frequency (*r_mating-illu_* = 0.473), followed by temperature (*r_mating-temp_* = 0.308) and humidity (*r_mating-humi_* = 0.308). Furthermore, illumination, temperature, and humidity were strongly associated with each other (*r_illu-temp_* = 0.872, *r_illu-humi_* = 0.614, *r_temp-humi_* = 0.726) (Table 1).

### 3.3. Effect of Adult Body Size on Mating Success

For *S. noctilio*, mating males were larger than unmated individuals with significant differences in three body characteristics. Mating females were larger than unmated individuals, but with no significant difference in terms of the three body characteristics.

For *S. nitobei*, mating males or females were larger than unmated individuals, but with no significant difference as to three body characteristics (Table 2).

### 3.4. Effect of Adult Age on Mating Frequency

Two-day-old females and males showed the strongest mating behavior in both species. When the males were maintained at 2-days-old by daily replacement of older individuals, the cumulative number of mating couples with 2-day-old females was highest (29.67 ± 3.06), with a significant difference between 2-day-old males and other ages, in *S. noctilio* (*p* = 0.001, *F* = 84.877, *df* = 6) (Figure 3a) and *S. nitobei* (20.33 ± 1.53 mating couples, *p* = 0.000, *F* = 71.773, *df* = 6) (Figure 3b). Similarly, when the age of the females was fixed at 2 days old, the cumulative number of mating couples with 2-day-old males was the highest (21.67 ± 4.73), with a significant difference between 2-day-old females and other ages, in *S. noctilio* (*p* = 0.000, *F* = 33.904, *df* = 6) (Figure 3a) and *S. nitobei* (17.33 ± 2.51 mating couples, *p* = 0.000, *F* = 46.715, *df* = 6) (Figure 3a). Compared with *S. nitobei,* the average mating frequency of *S. noctilio* was much higher at each tested age.

### 3.5. Effect of Sex Ratio on Daily Mating Frequency

The female-to-male ratio of 5:15 was the most conducive to mating for both species. When the number of females was kept constant and the number of males was increased, a female-to-male ratio of 5:15 achieved the highest mating frequency in *S. noctilio* (30.3 ± 6.1) and *S. nitobei* (12.0 ± 4.8) (Figure 4), and further increasing the number of males decreased the mating frequency. Compared with *S. nitobei,* the average mating frequency of *S. noctilio* was much higher at each female-to-male ratio.

### 3.6. Evidence of Male-Produced Pheromones in Two Woodwasps

Compounds in male SPME extracts were identified by comparing their retention times and mass spectra on DB-WAX (Figure 5) and HP-5 (Figure 6) with those of synthetic standards.

In the DB-WAX column (Figure 5), the mass spectrum of peak (a) from *S. noctilio* extract and peak (b) from *S. nitobei* extract exhibited the following peaks [*m/z* (relative abundance)]: 41, 55 (100, base), 68, 81, 95, 138, and 156. The mass spectrum had a molecular ion [M+] at m/z 156 and a significant [M+-18] ion at m/z 138 indicating loss of water, which was identical to that of peak (c) of synthetic standards (*Z*)-3-decen-1-ol. In addition, the retention times of peak (a) and (b) matched those of (*Z*)-3-decen-1-ol on the DB-WAX (20.396 min) (Figure 5a–c). Thus, the compound was identified as (*Z*)-3-decen-1-ol. Similarly, our results suggested that (*Z*)-3-decenol was also present in the SPME extracts of two populations. In HP-5 column (Figure 6a–c). However, two reported minor reference components, (Z)-4-decen-1-ol (Figure 5d and Figure 6d) and *(E, E*)-2,4-decadienal (Figure 5e and Figure 6e), were not identified in the SPME extract of either species on either type of column.

A similar pheromone release circadian rhythm was found for two species. Male pheromone release peak from 2 days of age occurred from 11:00 to 12:00, followed by 10:00 to 11:00 and 09:00 to 10:00 in male *S. noctilio* (Figure 7a, Table 3) and male *S. nitobei* (Figure 7b, Table 3). In addition, pheromone release was age dependent. Two-day-old males released the most (*Z*)-3-decenol, followed by three-day-old males in *S. noctilio* (Figure 8a, Table 4) and *S. nitobei* (Figure 8b, Table 4). For both species, mating behavior and male pheromone release were synchronized at different ages and times in the daytime.

## 4. Discussion

Observation of the mating behavior of woodwasps in their native forests is difficult because mating always occurs on the upper branches of trees. Here, the mating behavior of *S. nitobei* and *S. nitobei* was observed in outdoor cages. Compared with lepidopteron insects, these siricids did not show distinct calling behavior. In general, the mating behavior of *S. nitobei* was similar to that of *S. noctilio*: males approached females, touched antennae to find a suitable mating partner, and mated successfully. However, differences between these species were also observed. Compared with native species *S. nitobei,* the invasive species *S. noctilio* showed much stronger mating ability. Female and male *S. noctilio* cumulatively mated up to 19 and 13 times per day, respectively, and the average mating duration was 51.25 ± 25.54 s. However, female *S. nitobei* cumulatively mated up to 3 times per day, and males mated at most 4 times per day, while the average mating duration of the species was only 32.19 ± 16.02 s. The mating frequency of *S. noctilio* was much higher than that of *S. nitobei* regardless of the experimental context (circadian rhythm in a day, differences according to day, sex ratio).

Similar circadian rhythms were observed for both species, in which mating activity mainly occurred between 9:00 and 17:00 each day and peaked at 11:00 and 12:00. Caetano and Hajek [14] reported peak mating activity from 13:00 to 15:35. Another woodwasp, horntail *Sirex rufi-abdomins*, damages *P. massoniana* in Anhui province in southern China [21], and this species was found to mate mainly from 11:00 to 12:00. Taken together with these studies, our findings indicate that woodwasps generally mate around midday.

The mating behavior of both species was related to light intensity. Madden [1] reported that male *S. noctilio* swarmed near the bright tops of trees. Hurley et al. (2015) also reported that a clear intercept trap with light caught more female and male *S. noctilio* in comparison with a clear intercept without light. In our study, males gathered quickly in the upper corner of the cage, towards the sun, once they had been put into the cage. These findings suggest that *S. noctilio* and *S. nitobei* are photopositive. Temperature also influences siricid mating. Caetano and Hajek [13] reported that mating of *S. noctilio* mainly occurred in the brightest part of the cage, and the number of mating sessions was positively associated with temperature, which was in accordance with our results. Madden [1] found that the oviposition behavior of *S. noctilio* was positively correlated with temperature and peaked at 20–22 °C. Morgan [2] reported that the emergence behavior of adult *S. noctilio* was affected by weather conditions; most insects emerged on bright and sunny days, whereas fewer insects emerged on cloudy, humid days, and no adults emerged on rainy days. In summary, environmental conditions with appropriate light intensity and temperature are beneficial for siricid reproductive behavior.

Mating behavior was found to be age-dependent for both tested woodwasp species. *S. noctilio* and *S. nitobei* can mate soon after emergence; most females and males of both species mated at 2-days-old. Cooperband et al. [13] reported that male *S. noctilio* were only attracted to odors from 2–5-day-old males and not to those from 0–1-day-old males, which suggested that 2-day-old males released more pheromones. Many studies have reported that appropriate adult age is associated with stronger reproductive characteristics, including female fertility, egg hatching rate, and longevity [22,23,24,25].

The mating behavior of the tested species was also found to be related to the sex ratio. In our study, a female-to-male ratio of 5:15 was the most conducive to mating for both species, but mating success decreased when the number of males was increased to 20. These results are indicative of a male-biased species, which tend to follow particular reproductive strategies. Firstly, these species generally mate multiple times. In our study, 41.9% of *S. nitobei* females and 36.8% of *S. nitobei* males mated more than once in their lives. Campbell [26] indicated that the benefits of multiple mating by females exceeded the costs of refusing males, which was a waste of time and energy that also increased the risk of predation [27]. After multiple mating sessions, females received more sperm. Furthermore, females who mated multiple times lived longer and showed greater fertility [28]. Secondly, male-biased species generally have a relatively short mating duration. In our study, *S. nitobei* spent 8–73 s mating, with an average of 32.19 ± 16.24 s. Average mating duration in *S. noctilio* was 51.25 ± 25.54 s. Both mating durations were shorter than those of most insects, including hymenopterous parasitoid wasp *Campoletis chlorideae* (162.45 ± 9.4 s) [29], coleopterous *Monochamus alternatus* (63.49 min) [23], lepidopterous *Maruca vitrata* (75 min) (Lu et al. 2008). A short mating duration can increase mating success and reduce the risk of predation.

Mating behavior was also found to be related to body size. Fukuda et al. [30] showed that larger female adults were more likely to have reproductive success. Fukuda and Hijii [30] indicated that longevity and the number of eggs laid by female *S. nitobei* were positively correlated with fresh body weight, whereas male longevity was negatively correlated with fresh body weight. In our study, mated *S. noctilio* had longer bodies, longer proala, longer ovipositors, and wider thoraxes in comparison with unmated individuals.

In our study, (*Z*)-3-decenol was present in the SPME extracts of *S. noctilio* and *S. nitobei* populations. The morphology of *S. noctilio* and *S. nitobei* is very similar and they were reported to attack the same host, *P. sylvestris* var. *mongolica*, successively in Jinbaotun town, Inner Mongolia Autonomous Region, China. For the two sympatric species with similar pheromone components, what is the reproductive isolation mechanism? In Jinbaotun town, Inner Mongolia Autonomous Region, *S. noctilio* adults started to emerge in the field from late June to early September; subsequently, *S. nitobei* emerged from the same trees, with a peak emergence rate from late August through late September. Although the presence of the siricids in the region overlapped by a week or ten days at most, most of their development stage cannot meet in the field and they do not required special chemical cues to locate their own mates. However, several suspected minor components reported by Cooperband et al. [13], including (*Z*)-4-decen-1-ol) and *(E, E*)-2,4-decadienal, were not identified by GC-MS in extracts from *S. nitobei* or *S. noctilio*. Further study should be conducted to elucidate the crucial and minor pheromone components in both species.

Mating activity and male pheromone (*Z*)-3-decen-1-ol release were synchronized at different ages and times of the day in *S. noctilio*. In general, reproduction behavior coincides with pheromone release in insect species [24,31,32,33]. In many lepidopterous moths, sex pheromones are always released by females, and changes in male responsiveness and locomotor activity are observed accordingly [32,33,34]. The pheromone plume makes a powerful external stimulus to the receiver and could hypothetically function to synchronize rhythms in reproductive behaviors, which are primarily determined by rhythmic pheromone biosynthesis and endocrine system control [35]. This kind of synchronization may arise for both sexes through reliance on the same external zeitgebers, such as the photoperiod, temperature, and other environmental factors, as indicated by our finding that rhythmic mating frequency, pheromone release, and corresponding environmental factors are closely associated with different ages and times of the day.

## Figures and Tables

**Figure 1 insects-13-00966-f001:**
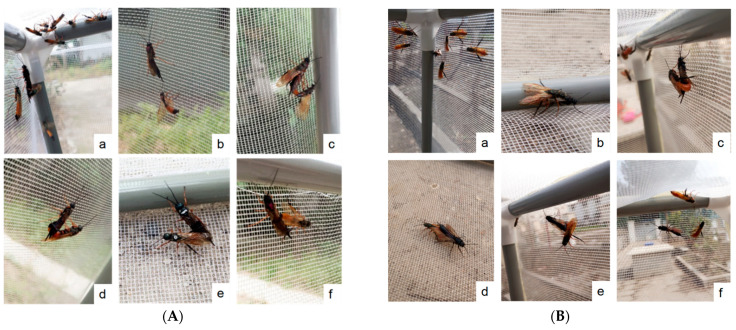
Calling and mating behavior of *Sirex noctilio* Fabricius (**A**) and *Sirex nitobei* Matsumura (**B**) a: Searching; b: Attracting; c: Seizing; d: Copulating on the cage wall; e: Copulating on the ground of the cage; f: Ending.

**Figure 2 insects-13-00966-f002:**
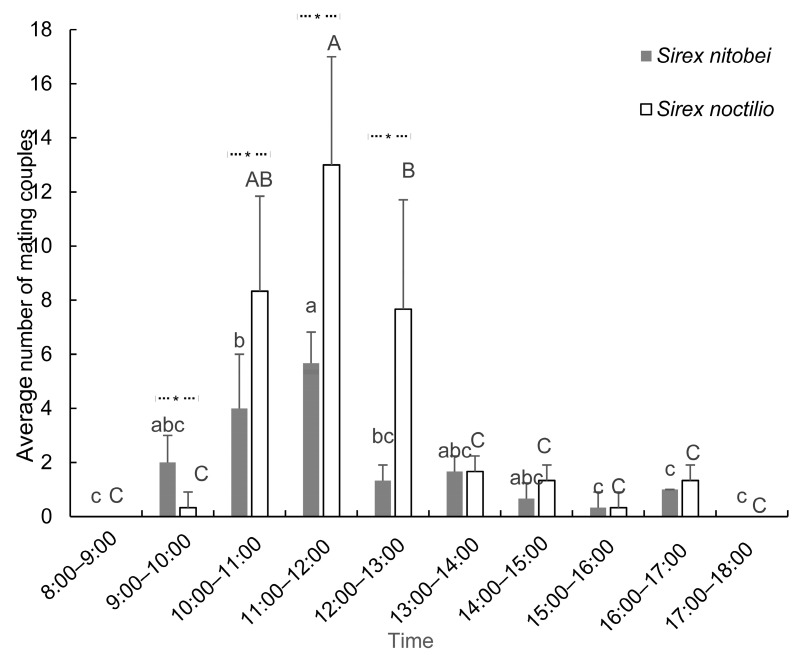
Mating circadian rhythms of two siricids. Different letters on the bars indicate significant differences (one-way ANOVA followed by Tukey’s multiple comparison test, *p* < 0.05). Significant differences between two siricids were analyzed by Mann–Whitney *U*-tests (*p* < 0.05; asterisked column, significant difference between two siricids).

**Figure 3 insects-13-00966-f003:**
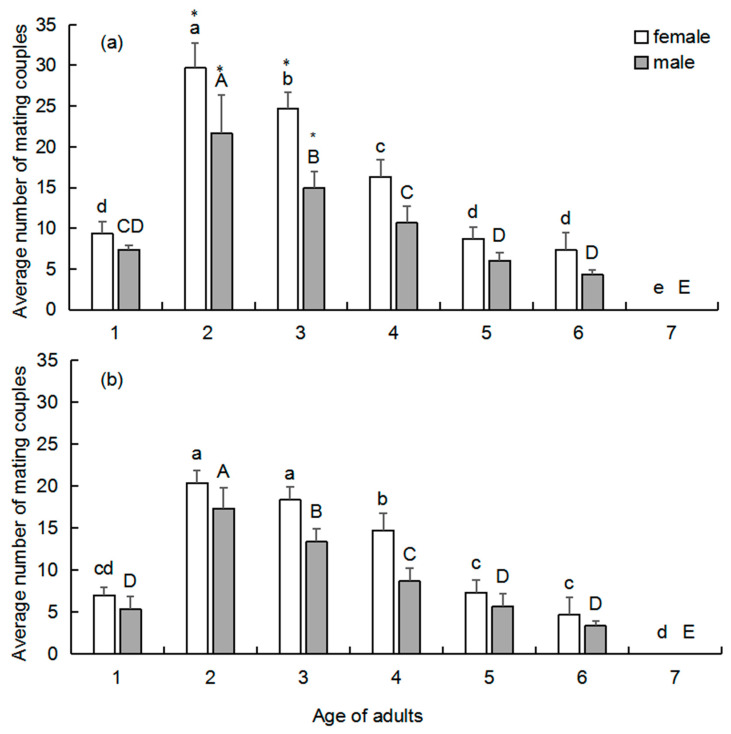
Effects of female and male age on mating frequency in *Sirex noctilio* (**a**) and *Sirex nitobei* (**b**). Different letters on the bars indicate significant differences (one-way ANOVA followed by Tukey’s multiple comparison test, *p <* 0.05). Significant differences between two siricids were analyzed by Mann–Whitney *U*-tests (*p* < 0.05; asterisked column, significant difference between two siricids).

**Figure 4 insects-13-00966-f004:**
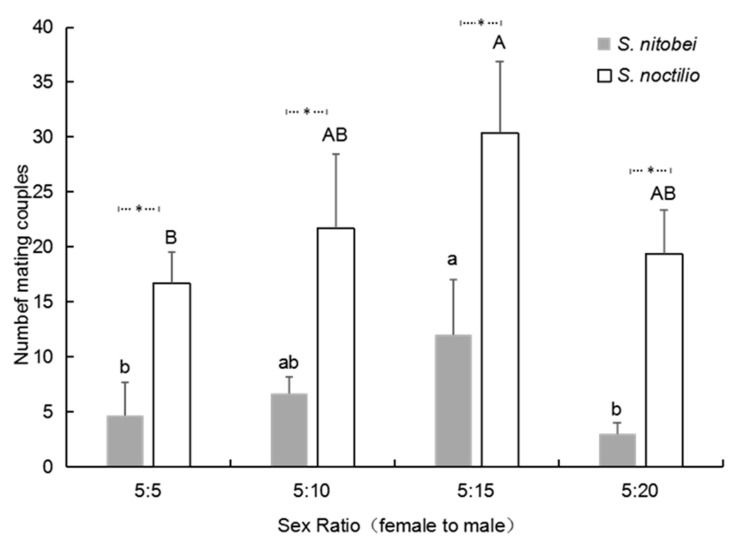
Effect of sex ratio on the daily mating frequency of two siricids. The experiment used 2-day-old adults in groups of 10 (female-to-male ratio, 5 to 5), 15 (female-to-male ratio, 5 to 10), 20 (female-to-male ratio, 5 to 15), or 25 (female-to-male ratio, 5 to 20). Different letters on the bars indicate significant differences (one-way ANOVA followed by Tukey’s multiple comparison test, *p* < 0.05). Significant differences between two siricids were analyzed by Mann–Whitney *U*-tests (*p* < 0.05; asterisked column, significant difference between two siricids).

**Figure 5 insects-13-00966-f005:**
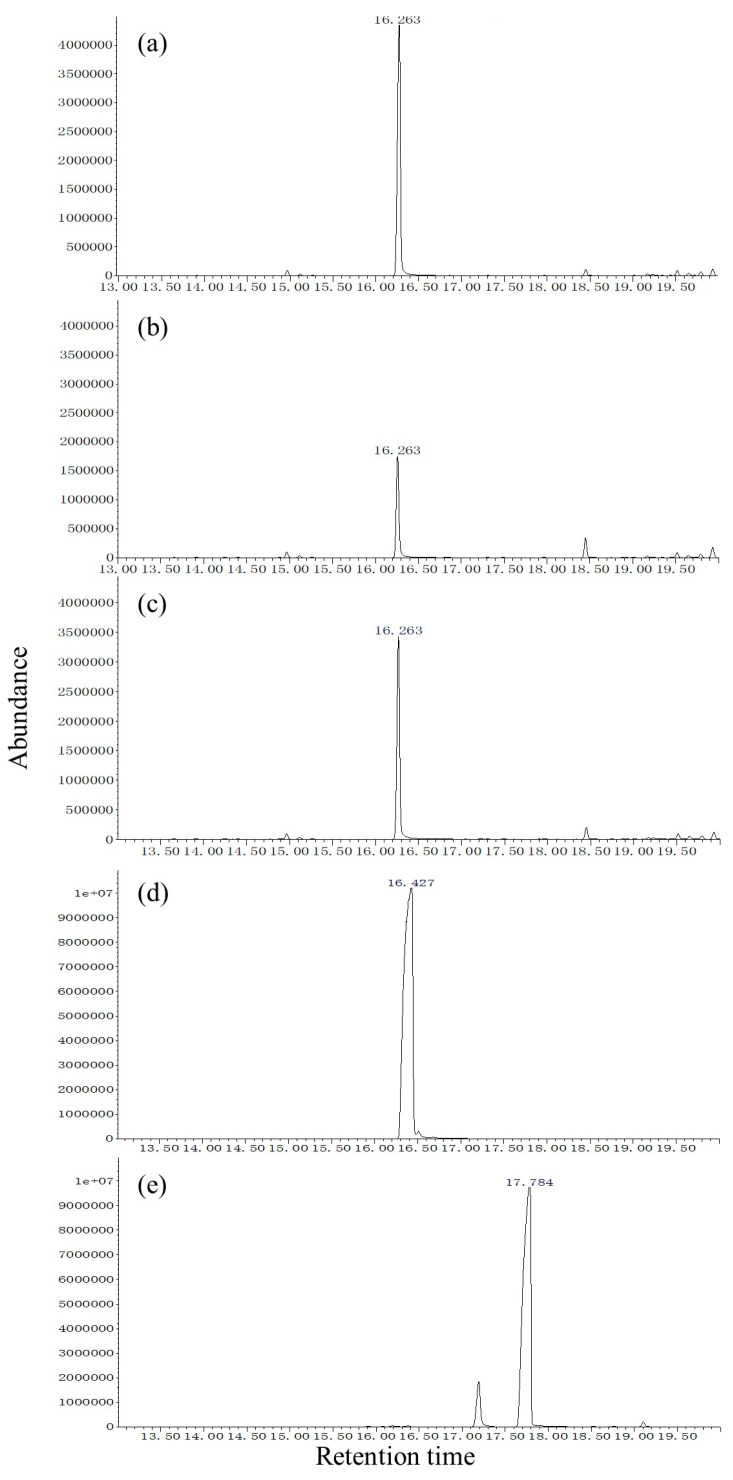
Total ion chromatograms from the GC-MS analysis of the SPME of *Sirex noctilio* extracts (**a**), *Sirex nitobei* extracts (**b**), (Z)-3-decen-1-ol (**c**), (Z)-4-decen-1-ol (**d**) and (E, E)-2,4-decadienal (**e**) with a DB-WAX column The temperature program was as follows: 50 °C for 1 min, then 5 °C/min to 240 °C and maintained for 10 min. MS conditions: interface temperature, 250 °C; ion source temperature, 250 °C; ionization voltage, 70 eV. nE+07 means n × 10^7^.

**Figure 6 insects-13-00966-f006:**
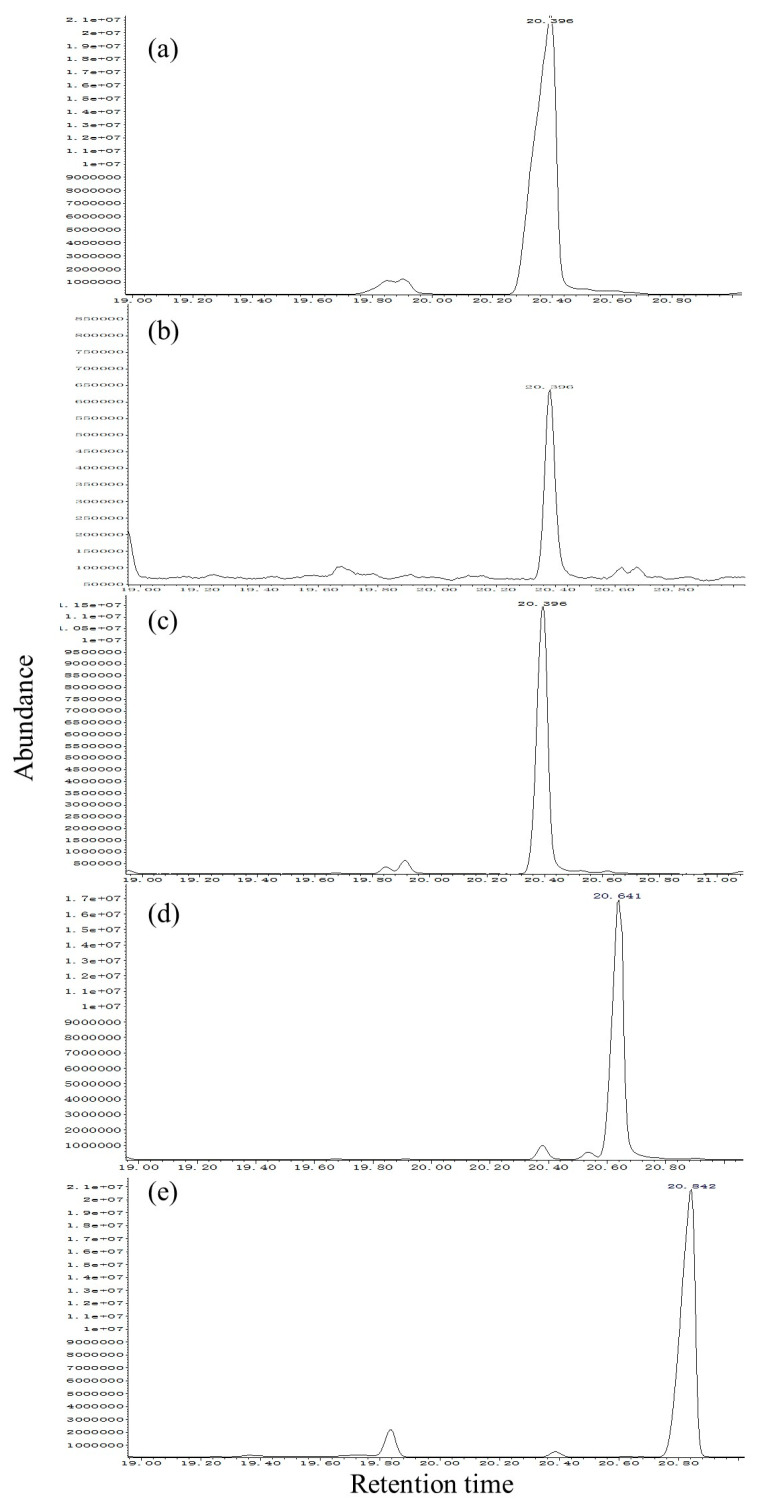
Total ion chromatograms from the GC-MS analysis of the SPME of *Sirex noctilio* extracts (**a**), *Sirex nitobei* extracts (**b**), (Z)-3-Decen-1-ol (**c**), (Z)-4-decen-1-ol (**d**), and (E, E)-2,4-decadienal (**e**) with a HP-5 column. The temperature program was as follows: 40 °C for 2 min, then 6 °C/min to 180 °C and 15 °C/min to 270 °C. MS conditions: interface temperature, 250 °C; ion source temperature, 250 °C; ionization voltage, 70 eV. nE+07 means n × 10^7^.

**Figure 7 insects-13-00966-f007:**
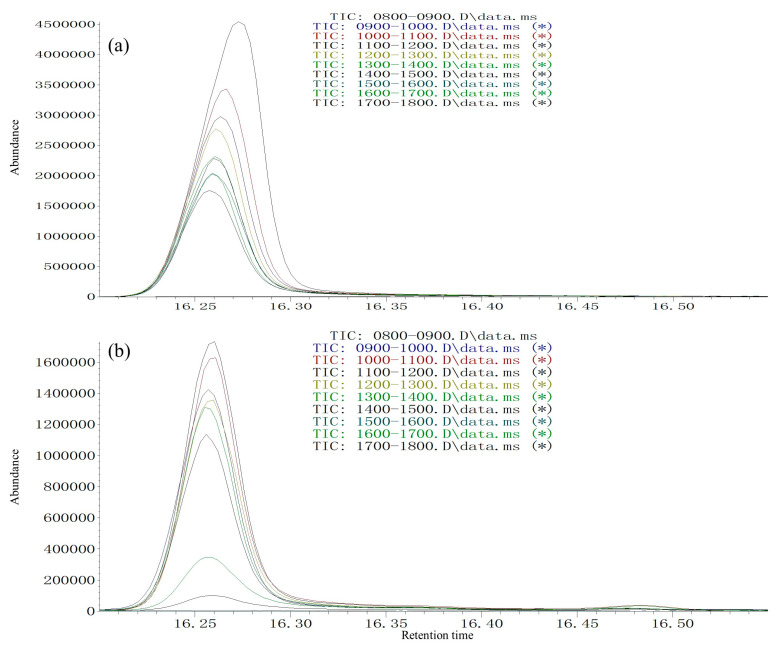
(A) Circadian (Z)-3-Decen-1-ol release collected from 8:00 to 18:00 from 2-day-old males of *S**irex*
*noctilio* (**a**) and *Sirex nitobei* (**b**); GC conditions: HP-5 fused-silica column, 40 °C for 2 min, then 6 °C/min to 180 °C and 15 °C/min to 270 °C. MS conditions: interface temperature, 250 °C; ion source temperature, 250 °C; ionization voltage, 70 eV.

**Figure 8 insects-13-00966-f008:**
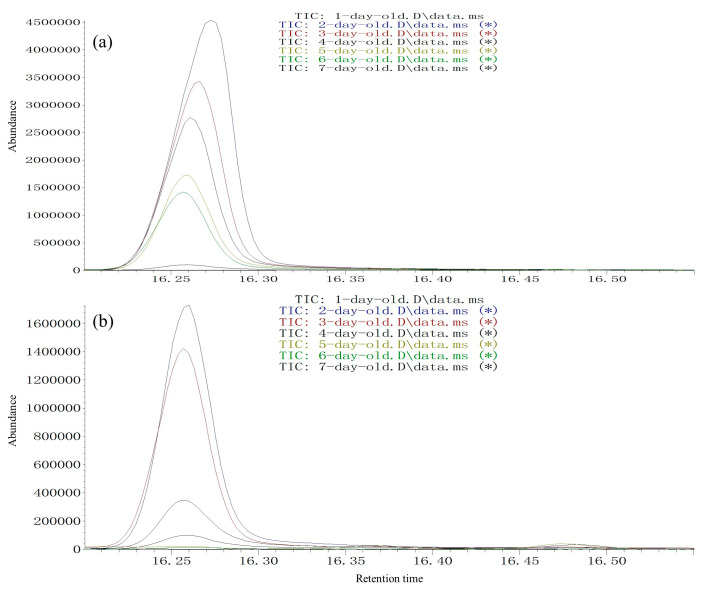
Age-dependent (*Z*)-3-Decen-1-ol release collected from 11:00–12:00 from 2- to 7-day-old males from *Sirex noctilio* (**a**) and *Sirex nitobei* (**b**); GC conditions: HP-5 fused-silica column, 40 °C for 2 min, then, 6 °C/min to 180 °C and 15 °C/min to 270 °C. MS conditions: interface temperature, 250 °C; ion source temperature, 250 °C; ionization voltage, 70 eV.

**Table 1 insects-13-00966-t001:** The temperature, humidity, and illumination at different times.

Time	Temperature (°C) ^1^	Humidity (%) ^1^	Illumination (lux) ^1^
08:00–09:00	27.0 ± 6.0 ^a^	50.0 ± 17.3 ^a^	39,000.0 ± 29,901.3 ^abc^
09:00–10:00	30.0 ± 8.0 ^a^	43.0 ± 22.5 ^a^	52,200.0 ± 28,754.8 ^abc^
10:00–11:00	31.7 ± 10.0 ^a^	40.7 ± 23.1 ^a^	54,100.0 ± 35,024.7 ^abc^
11:00–12:00	35.0 ± 3.0 ^a^	26.7 ± 11.5 ^a^	66,800.0 ± 12,646.3 ^a^
12:00–13:00	33.2 ± 9.5 ^a^	38.0 ± 19.6 ^a^	59,966.7 ± 29,876.0 ^ab^
13:00–14:00	33.8 ± 2.5 ^a^	30.7 ± 11.0 ^a^	50,033.3 ± 8808.1 ^abc^
14:00–15:00	31.5 ± 4.3 ^a^	32.0 ± 11.0 ^a^	31,233.3 ± 15,580.9 ^abc^
15:00–16:00	28.7 ± 4.2 ^a^	41.7 ± 17.9 ^a^	20,900.0 ± 8524.7 ^bc^
16:00–17:00	24.5 ± 2.2 ^a^	45.3 ± 10.4 ^a^	11,233.3 ± 6149.3 ^c^
17:00–18:00	21.3 ± 1.2 ^a^	54.3 ± 9.3 ^a^	9875.3 ± 4568.2 ^c^

^1^ Numbers in the same column with the same letters are not significantly different according to one-way ANOVA followed Tukey’s multiple range test at a level of 5%.

**Table 2 insects-13-00966-t002:** Comparison on body characteristics of mated and non-mated individuals of two Siricids.

Sex	Siricidaes	Mated or Not	Body Length (mm) ^1^		Pronotum (mm) ^1^		Proala Length (mm) ^1^		Ovipositor Length (mm) ^1^	
Male	*Sirex noctilio*	Yes	18.18 ± 3.26 *	*p* = 0.010	2.94 ± 0.62 *	*p* = 0.022	13.16 ± 2.41 *	*p* = 0.019	—	—
		No	16.39 ± 3.13	*t* = 2.615	2.64 ± 0.54	*t* = 2.332	12.00 ± 1.99	*t* = 2.373	—	—
	*Sirex nitobei*	Yes	16.52 ± 2.09	*p* = 0.434	2.88 ± 0.43	*p* = 0.836	12.63 ± 1.66	*p* = 0.314	—	—
		No	16.03 ± 2.73	*t* = −0.787	2.83 ± 0.52	*t* = 0.208	12.22 ± 1.57	*t* = −1.104	—	—
Female	*Sirex noctilio*	Yes	20.14 ± 3.57	*p* = 0.017	3.11 ± 0.59	*p* = 0.227	14.47 ± 2.20	*p* = 0.067	10.41 ± 1.56	*p* = 0.168
		No	17.89 ± 3.64	*t* = 1.337	2.77 ± 0.60	*t* = 1.223	12.57 ± 1.61	*t* = 1.870	9.36 ± 1.93	*t* = 1.397
	*SIrex nitobei*	Yes	17.31 ± 3.08	*p* = 0.256	3.02 ± 0.61	*p* = 0.122	13.27 ± 1.93	*p* = 0.188	9.60 ± 1.33	*p* = 0.231
		No	16.09 ± 2.28	*t* = 1.154	2.7 ± 0.37	*t* = 1.579	12.39 ± 1.28	*t* = 1.340	9.04 ± 1.08	*t* = 1.216

^1^ Student’s paired-sample *t*-test was used to identify significant differences between body size of mated and unmated siricids. * Indicates that there is a significant difference between mated and unmated male *S. noctilio* in body Length, pronotum, and proala length (*p* < 0.05).

**Table 3 insects-13-00966-t003:** Relative amount of (*Z*)-3-decenol in male siricids SPME extracts collected from 8:00 to 18:00 at 2 days of age.

Time	Relative Amount of (*Z*)-3-Decenol (Mean+ SD) ^1^	
	*Sirex noctilio*	*Sirex nitobei*
08:00–09:00	-	-
09:00–10:00	62.82 ± 1.69 ^c^	79.68 ± 0.91 ^c^
10:00–11:00	71.66 ± 1.89 ^b^	94.51 ± 2.12 ^b^
11:00–12:00	100.0 ± 0.00 ^a^	100.0 ± 0.00 ^a^
12:00–13:00	54.36 ± 1.14 ^d^	79.97 ± 1.39 ^c^
13:00–14:00	45.78 ± 1.21 ^e^	72.07 ± 3.35 ^d^
14:00–15:00	44.48 ± 0.81 ^ef^	63.71 ± 0.61 ^e^
15:00–16:00	41.69 ± 0.24 ^fg^	-
16:00–17:00	39.85 ± 0.40 ^g^	-
17:00–18:00	34.39 ± 1.55 ^h^	-

^1^ Three experimental replicates (15 × 3 males) were used for each of the ten sampling time-points. Numbers in the same column with the same letters are not significantly different according to one-way ANOVA followed Tukey’s multiple range test at a level of 5%. Quantities expressed relative to the most abundant time point (set to a value of 100) in the two species.

**Table 4 insects-13-00966-t004:** Relative amount of (*Z*)-3-decenol in male siricid SPME extracts collected during the 11:00–12:00 period from 2 to 7 days of age.

Ages	Relative Amount of (*Z*)-3-Decenol (Mean+ SD) ^1^	
	*Sirex noctilio*	*Sirex. nitobei*
1-day-old	-	-
2-day-old	100.0 ± 0.00 ^a^	100.0 ± 0.00 ^a^
3-day-old	71.69 ± 2.19 ^b^	79.02 ± 1.76 ^b^
4-day-old	55.09 ± 0.69 ^c^	20.23 ± 0.38 ^c^
5-day-old	33.29 ± 1.23 ^d^	-
6-day-old	26.56 ± 1.05 ^e^	-
7-day-old	-	-

^1^ Three experimental replicates (15*3 males) were used for each of the seven sampling ages. Numbers in the same column with the same letters are not significantly different according to one-way ANOVA followed by Tukey’s multiple range test at a level of 5%. Quantities expressed relative to the most abundant ages point (set to a value of 100) in the two species.

## Data Availability

The data presented in this study are available in article.

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
