# Peer review of "Mating Behavior and Identification of Male-Produced Pheromone Components in Two Woodwasps, Sirex noctilio and Sirex nitobei, in China"

_insects, 2022, doi:10.3390/insects13100966_

Round 1

Reviewer 1 Report

This is a well written manuscript that describes the mating behavior of two wood wasp species found in China.  Mating behaviors were followed in artificial cages exposed to various conditions.  Only one volatile component was found.  Based on the method of detecting the compound and mating behavior observations it is unclear if this compound is a pheromone.  The mating behavior assay did not demonstrate that a volatile chemical was used in communication. Instead it seems that nonvolatile cues are important due to the sequence of events used during mating. This should be discussed in context of the mating sequence where males are touching the females.  A contact pheromone may be present for mating to continue after the first contact with the antennae.  It is not clear if using Z3-decenol in a trap would attract female siricids.  A few specific comments are listed below.

Line 141 – Were adults fed anything or given water prior to and during the mating experiments?

Line 178 – Indicate how long the mating pair remained attached.  If it was less than 20 minutes then some mating behavior could have been missed.  An average mating duration of 51 seconds would miss a lot of mating behavior.

Line 221 – A standard curve should have made using known amounts of Z3-decenol inside an empty flask and the same SPME fiber used for calibration.

Line 314 – Table 2 needs a heading.

Line 382 – It is unclear what is shown in Figure 7. The figure legend was missing.

Author Response

Thank you very much for your patient review and kind suggestions.Please check our itemized responses below (file Review_Report1.docx) and our revisions in re-submitted MS (Revisions in MS were highlighted in red).

Reviewer 2 Report

The manuscript by Lu et al. on mating in two species of Sirex and their male pheromones is well written, and adds some novel aspects to the pheromonal biology of Sirex, but it is not a major breakthrough. The components the authors are dealing with have already been described, and were re-identified for Sirex nitobei. The evaluation of a rhythm of pheromone release which corresponds to mating is the most interesting part of this study, but particularly this part has some methodological shortcomings. The quantification of pheromone release is done by SPME, which is a method generally not suited to quantify components since it is a non-exhaustive extraction technique. At least, the authors should repeat these experiments using a reference quantification curve with known amounts of the synthetic standards or better internal standards. I am thus afraid that all quantifications presented in the manuscript (and these represent the core of the whole study!) are not reliable.

Apart from this major shortcoming, there are several minor issues that need to be fixed. For instance, the authors use two chromatographic columns, a DB-WAX and a HP-5. A look at figures 5 and 6 clearly shows that (Z)-3-decen-1-ol and (Z)-4-decen-1-ol are only separable on the DB-WAX column, but have the same retention on the HP-5 column. I do not understand why this second column has been used at all and why the authors present a large composed figure on an issue that did not work.

There are a few further minor issues, mainly regarding style and presentation. For instance, Figure 7 has no specific legend but is just a placeholder with a generalized text not related to the figure. Furthermore, instead of retention times for compounds, the authors should provide retention indices – this is important for a comparison to literature data.

I recommend a major revision.

Author Response

Thank you very much for your patient review and kind suggestions.Please check our itemized responses below (file Review_Report2.docx) and our revisions in re-submitted MS (Revisions in MS were highlighted in red).

Round 2

Reviewer 1 Report

In my opinion the authors have made sufficient arguments and corrections for publication.